# TransBox: $\mathcal{EL}^{++}$-closed Ontology Embedding

## Abstract

OWL (Web Ontology Language) ontologies, which are able to represent both relational and type facts as standard knowledge graphs and complex domain knowledge in Description Logic (DL) axioms, are widely adopted in domains such as healthcare and bioinformatics. Inspired by the success of knowledge graph embeddings, embedding OWL ontologies has gained significant attention in recent years. Current methods primarily focus on learning embeddings for atomic concepts and roles, enabling the evaluation based on normalized axioms through specially designed score functions. However, they often neglect the embedding of complex concepts, making it difficult to infer with more intricate axioms. This limitation reduces their effectiveness in advanced reasoning tasks, such as Ontology Learning and ontology-mediated Query Answering. In this paper, we propose $\mathcal{EL}^{++}$-closed ontology embeddings which are able to represent any logical expressions in DL $\mathcal{EL}^{++}$ via composition. Furthermore, we develop TransBox, an effective $\mathcal{EL}^{++}$-closed ontology embedding method that can handle many-to-one, one-to-many and many-to-many relations. Our extensive experiments demonstrate that TransBox often achieves state-of-the-art performance across various real-world datasets for predicting complex axioms. Code and data are available at the https://anonymous.4open.science/r/TransBox-F4B7.

## Keywords

Ontology Embedding, Description Logic, Web Ontology Language, Ontology Completion, Ontology Learning

## 1 Introduction

Ontologies[1] structured according to the Web Ontology Language (OWL) standards developed by the W3C [14] are extensively used across various domains such as the Semantic Web [16], healthcare [13], finance [4], and biology [6]. In recent years, OWL ontology embeddings, which are vector-based knowledge presentations, have gained significant attention due to the increasing need for more effective methods to predict or infer missing knowledge as well as for more wider ontology application especially in combination with machine learning [11].

Unlike Knowledge Graphs [18], which focus on individual entities (a.k.a. instances), ontologies emphasize concepts that represent groups of individuals, allowing for richer semantic interpretations. These interpretations are typically formalized through Description Logics (DLs) [3], which provide a logical foundation for reasoning within ontologies.

In ontologies, knowledge is often represented and leveraged through the intricate composition of atomic concepts and roles. For example, defining a disease typically requires the integration of various components, such as its symptoms and the affected locations in the body (Formally, Disease ≡ ∃*hasSymptom*.Symptom ⊓

---

[1]This study explores advanced methods related to OWL ontologies, a widely used Web-based approach for knowledge representation, and aligns with the focus of the semantics and knowledge track at the Web Conference.

∃*occursIn*.BodyLocation ⊓ . . .). Similarly, when utilizing this information for diagnosis, one must consider a wide array of patient data (see Section 6.2.4 for a more concrete example). Embedding methods must effectively capture these complex constructs, which are formed by logical operators (e.g., conjunction ⊓, existential quantification ∃*r*), to accurately model intricate knowledge and enhance practical applicability in real-world scenarios. This capability is essential for a variety of applications, such as Ontology Learning [23], ontology-mediated query answering [5], and tasks like updating ontologies to incorporate newly emerging complex concepts from external resources, such as the latest clinical guidelines or research findings [12].

In this work, we propose $\mathcal{EL}^{++}$-closed ontology embeddings that guarantee all logical operations in $\mathcal{EL}^{++}$ (i.e., ⊓, ∃*r*, and role composition ◦) are effectively captured. In other words, $\mathcal{EL}^{++}$-closed embedding could generate the embedding of any complex $\mathcal{EL}^{++}$ concept by composing the embeddings of atomic concepts, and thus could be applied in many ontology reasoning tasks beyond the standard subsumption predictions over atomic concepts. However, we show that most existing methods, whether based on language models (LMs) or geometric models, are either not $\mathcal{EL}^{++}$-closed or have other theoretical limitations.

Firstly, LM-based approaches [9, 10, 28] rely on textual information, such as concept descriptions, to predict logical relationships. However, these approaches obscure the reasoning process within large neural networks and are unable to guarantee adherence to any formal logical constraint, including $\mathcal{EL}^{++}$-closeness.

Secondly, geometric model-based methods also face challenges related to $\mathcal{EL}^{++}$-closure or theoretical limitations. For example, methods that embed concepts as balls, such as ELEM [22] and EMEM++ [24], struggle to handle conjunctions (⊓) because the intersection of two balls is generally not a ball.

On the other hand, methods that embed concepts as boxes, such as BoxEL [31], ELBE [26], and Box$^2$EL [17], generally maintain closure under ⊓, except Box$^2$EL, which introduces bump vectors that are not defined for conjunctions like $A \sqcap B$. Despite being more effective with conjunctions, box embedding methods face several significant challenges, as outlined below:

- *Inadequate handling of roles.* BoxEL and ELBE embed roles as invertible mappings, inherently assuming that all roles are functional across instances. This assumption limits their ability to handle many-to-one, one-to-many, and many-to-many roles, making it challenging to accurately embed even simple ontologies (an example is deferred to Section 6.1). In contrast, Box$^2$EL represents roles as products of boxes, allowing it to effectively capture many-to-many relationships. However, it faces difficulties with role composition, as all roles are implicitly treated as transitive (see Appendix B).

- *Issues with box intersections.* In relatively low-dimensional spaces, such as $\mathbb{R}^{50}$, box embeddings often struggle to produce non-empty intersections, as outlined in Theorem 4.5. Consequently, this may result in disjoint boxes where overlap is expected. This

limitation undermines the model's ability to accurately represent complex concepts that involve conjunctions of parts and restricts the scalability of the embedding methods to higher-dimensional spaces.

To address these challenges, we propose a novel $\mathcal{EL}^{++}$-closed ontology embedding method, *TransBox*. **Firstly**, TransBox captures many-to-many roles by embedding roles as group transitions of vectors within a box, rather than relying on the single transition used in previous methods. This approach effectively models more complex relational dynamics and provides a natural mechanism for capturing role compositions. **Secondly**, TransBox introduces a novel approach to handling box intersections. It defines intersections using the space $(\mathbb{R} \cup \emptyset)^n$ rather than $\mathbb{R}^n$, thus offering more flexibility in embeddings. This extension enables the model to effectively handle scenarios where non-empty overlaps between concept boxes are expected.

Our main contributions can be summarized as follows:

- We propose $\mathcal{EL}^{++}$-closed ontology embeddings that can represent all logical information expressed in the description logic $\mathcal{EL}^{++}$, including role compositions.
- We introduce TransBox, an $\mathcal{EL}^{++}$-closed embedding method that effectively handles the embedding of many-to-many roles, role compositions, and overlapping concepts, and has been proved to be sound.
- Experimental results on three real-world ontologies demonstrate that our method often achieves state-of-the-art performance in predicting complex axioms.

## 2 Preliminaries and Related Work

### 2.1 Ontologies

Ontologies use sets of statements (axioms) about concepts (unary predicates) and roles (binary predicates) for knowledge representation and reasoning. We focus on $\mathcal{EL}^{++}$-ontologies which keep a good balance between expressivity and reasoning efficiency with wide application [2]. Let $N_C = \{A, B, \ldots\}$, $N_R = \{r, t, \ldots\}$, and $N_I = \{a, b, \ldots\}$ be pair-wise disjoint sets of *concept names* (also called *atomic concepts*) and *role names*, and *individual names*, respectively. $\mathcal{EL}^{++}$-*concepts* are recursively defined from atomic concepts, roles and individuals as

$$\top \mid \perp \mid A \mid C \sqcap D \mid \exists r.C \mid \{a\} \qquad (1)$$

and an $\mathcal{EL}^{++}$-*ontology* is a finite set of TBox axioms of the form

$$C \sqsubseteq D, \quad r \sqsubseteq t, \quad r_1 \circ r_2 \sqsubseteq t$$

and ABox axioms $A(a)$ or $r(a, b)$. Note $C, D$ are (possibly complex) $\mathcal{EL}^{++}$-concepts, $A$ is a concept name, $r_1, r_2, t$ are role names, and $a, b$ are individual names. The following is an example of axioms of a toy family ontology (see Figure 5 for the complete version).

EXAMPLE 1. *Using atomic concepts Father, Child, Male, . . ., role hasParent, and individuals Tom, Jerry, we can construct a small family ontology consisting of two TBox axioms*

$$Father \sqsubseteq Male \sqcap Parent, \quad Child \sqsubseteq \exists hasParent.Father,$$

*and two ABox axioms: Father(Tom), hasParent(Jerry, Tom).*

An *interpretation* $\mathcal{I} = (\Delta^\mathcal{I}, \cdot^\mathcal{I})$ consists of a non-empty set $\Delta^\mathcal{I}$ and a function $\cdot^\mathcal{I}$ mapping each $A \in N_C$ to $A^\mathcal{I} \subseteq \Delta^\mathcal{I}$, each $r \in N_R$ to $r^\mathcal{I} \subseteq \Delta^\mathcal{I} \times \Delta^\mathcal{I}$, and each $a \in N_I$ to $a^\mathcal{I} \in \Delta^\mathcal{I}$, where $\perp^\mathcal{I} = \emptyset$, $\top^\mathcal{I} = \Delta^\mathcal{I}$, $\{a\}^\mathcal{I} = a^\mathcal{I}$. The function $\cdot^\mathcal{I}$ is extended to any $\mathcal{EL}^{++}$-concepts by:

$$(C \sqcap D)^\mathcal{I} = C^\mathcal{I} \cap D^\mathcal{I}, \qquad (2)$$

$$(\exists r.C)^\mathcal{I} = \left\{ a \in \Delta^\mathcal{I} \mid \exists b \in C^\mathcal{I} : (a, b) \in r^\mathcal{I} \right\}, \qquad (3)$$

$$(r_1 \circ r_2)^\mathcal{I} = \left\{ (a, c) \mid \exists b \in \Delta^\mathcal{I} : (a, b) \in r_1^\mathcal{I}, (b, c) \in r_2^\mathcal{I} \right\}. \qquad (4)$$

An interpretation $\mathcal{I}$ *satisfies* a TBox axiom $X \sqsubseteq Y$ if $X^\mathcal{I} \subseteq Y^\mathcal{I}$ for $X, Y$ being two concepts or two role names, or $X$ being a role chain and $Y$ being a role name. It satisfies an ABox axiom $A(a)$ if $a^\mathcal{I} \in A^\mathcal{I}$ and it satisfies an ABox axiom $r(a, b)$ if $(a^\mathcal{I}, b^\mathcal{I}) \in r^\mathcal{I}$. Finally, $\mathcal{I}$ is a *model* of $O$ if it satisfies every axiom in $O$. An ontology $O$ *entails* an axiom $\alpha$, written $O \models \alpha$ if $\alpha$ is satisfied by all models of $O$.

### 2.2 Ontology Embeddings

There are two primary kinds of ontology embedding approaches: those based on Language Models (LMs) and those based on geometric models. LM-based approaches, including those based on traditional non-contextual word embedding models and those based on Transformer [30] with contextual embeddings, such as OPA2Vec [28], OWL2Vec [10], and BERTSub [9], rely on textual data (e.g., concept descriptions) and predict logical informations by LMs. However, these methods loosely capture the underlying conceptual structure and often lack the interpretability needed for human understanding. In this work, we focus on geometric model-based approaches, which offer more intuitive representations.

Geometric model-based methods use different geometric objects for constructing the geometric models of ontologies. For example, cones [15, 33] and fuzzy sets [29] have been used for $\mathcal{ALC}$-ontologies. For $\mathcal{EL}$-family ontologies, most methods using either boxes (Box²EL [17], BoxEL [31], ELBE [26]) or balls (ELEM [22], EMEM++ [24]). Among these, box-based methods have gained popularity due to their closure under intersection, which aligns well with the intersection of concepts in Description Logic. In contrast, ball-based embeddings are not closed under intersection, as the intersection of balls does not typically form a ball.

However, these existing methods only consider embedding atomic concepts but neglect the embeddings of complex concepts, which prevents them from evaluating axioms beyond normalized ones. This limitation restricts their applicability to tasks involving complex axioms. In this work, we propose the $\mathcal{EL}^{++}$-closed ontology embedding, which is able to generate embeddings of complex concepts from atomic ones, allowing for both training and evaluation across a wider variety of ontologies. Our analysis reveals that a large part of the existing geometric model-based approaches are not $\mathcal{EL}^{++}$-closed, while the remaining methods cannot model many-to-many relationships, demonstrating suboptimal performance when tested on real-world datasets.

## 3 $\mathcal{EL}^{++}$-Closed Embeddings

Before introducing our ontology embedding method, we first present the concept of $\mathcal{EL}^{++}$-closed embedding. Given an ontology $O$, the goal of ontology embedding is to create a geometric model of the ontology $O$ that "faithfully" represents the meaning of concepts,

roles, and individuals in $O$. In this model, each element in $\mathsf{N_C}$, $\mathsf{N_R}$, and $\mathsf{N_I}$ occurring in $O$ is mapped to a geometric object in an embedding space $\mathbb{S}$ (*e.g.*, linear space $\mathbb{R}^n$). For clarity, we define:

- $\mathcal{E}_C$: the set of all geometric objects, such as balls or boxes, that are considered as subsets of $\mathbb{S}$ and serve as candidates for concept embedding.
- $\mathcal{E}_R$: the set of all geometric objects that are considered as subsets of $\mathbb{S} \times \mathbb{S}$ and serve as candidates for role embedding.

To align with the machine learning framework, the elements of $\mathcal{E}_C$ and $\mathcal{E}_R$ should have numerical representations by vectors or specific values. We omit the collection for individuals because individuals are often embedded as points in the space $\mathbb{S}$, and therefore the collection is equivalent to the space itself.

Example 2. *Different embedding methods define their ontology embedding spaces in distinct ways. For example:*

- *For ELEM and ELEM++, $\mathcal{E}_C$ consists of all n-dimensional balls represented by their centers in $\mathbb{R}^n$ and radii in $\mathbb{R}$, and $\mathcal{E}_R$ consists of subsets defined by vector-based translations: $E_{\mathbf{v}_r} = \{(\mathbf{x}, \mathbf{x} + \mathbf{v}_r) \mid \mathbf{x} \in \mathbb{R}^n\}$, where $\mathbf{v}_r \in \mathbb{R}^n$.*
- *For Box$^2$EL, when all the bump vectors are set to zero vector, $\mathcal{E}_C$ consists of n-dimensional boxes[2] represented by their lower left and upper right corners in $\mathbb{R}^n$ or by their centers and offsets both in $\mathbb{R}^n$, and $\mathcal{E}_R$ consists of products of boxes, $Head(r) \times Tail(r) \subseteq \mathbb{R}^n \times \mathbb{R}^n$.*
- *For ELBE, $\mathcal{E}_C$ also consists of boxes as in Box$^2$EL, and $\mathcal{E}_R$ is defined similarly to the ELEM case.*
- *For BoxEL, $\mathcal{E}_C$ consists of boxes as in Box$^2$EL and ELBE, and $\mathcal{E}_R$ consists of subsets defined by affine transformations: $E_{(\mathbf{k}_r, \mathbf{b}_r)} = \{(\mathbf{x}, \mathbf{k}_r \cdot \mathbf{x} + \mathbf{b}_r) \mid \mathbf{x} \in \mathbb{R}^n\}$, where $\mathbf{k}_r, \mathbf{b}_r \in \mathbb{R}^n$.*

The $\mathcal{EL}^{++}$-closed embeddings refer to embedding methods where the assigned spaces $\mathcal{E}_C$ (for concepts) and $\mathcal{E}_R$ (for roles) are closed under the $\mathcal{EL}^{++}$-semantics defined by Equations 2, 16, and 4. The formal definition is provided in Definition 3.1, where each of the three conditions directly corresponds to these equations.

Definition 3.1 ($\mathcal{EL}^{++}$-*Closed Embeddings*). An ontology embedding method over a space $\mathbb{S}$, with embedding candidate sets $\mathcal{E}_C$ for concepts and $\mathcal{E}_R$ for roles, is $\mathcal{EL}^{++}$-*closed* if $\mathcal{E}_C$ includes the empty set $\emptyset$, the whole space $\mathbb{S}$, all singletons $\{\mathbf{x}\}$ with $\mathbf{x} \in \mathbb{S}$, and satisfies the following closure properties:

(1) *Conjunction closure*: $S \cap S' \in \mathcal{E}_C$ for all $S, S' \in \mathcal{E}_C$;
(2) *Existential quantification closure*: $\exists_E S \in \mathcal{E}_C$, where $\exists_E S = \{\mathbf{x} \mid \exists \mathbf{y} \in \mathbb{S} : (\mathbf{x}, \mathbf{y}) \in E\}$ for any $E \in \mathcal{E}_R$ and $S \in \mathcal{E}_C$;
(3) *Role composition closure*: $E \circ E' \in \mathcal{E}_R$ for all $E, E' \in \mathcal{E}_R$, where $E \circ E' = \{(\mathbf{x}, \mathbf{z}) \mid \exists \mathbf{y} \in \mathbb{S} : (\mathbf{x}, \mathbf{y}) \in E \text{ and } (\mathbf{y}, \mathbf{z}) \in E'\}$.

We require that $\mathcal{E}_C$ includes $\emptyset$, $\mathbb{S}$, and $\{\mathbf{x}\}$ to capture the semantics of $\bot$, $\top$, and $\{a\}$, respectively. It is important to note that this closure property is not specific to any particular ontology; rather, it is an inherent characteristic of the ontology embedding method itself.

Example 3. *ELEM and ELEM++ are not $\mathcal{EL}^{++}$-closed, as the collection of n-balls violates the first condition. Box$^2$EL also violates the first requirement, as bump vectors cannot be defined for any conjunction. In contrast, it can be verified that ELBE and BoxEL are $\mathcal{EL}^{++}$-closed.*

[2]In the general case, $\mathcal{E}_C$ consists of products of the form $Box \times \{\mathbf{v}\}$, where $\mathbf{v} \in \mathbb{R}^n$ represents bump vectors.

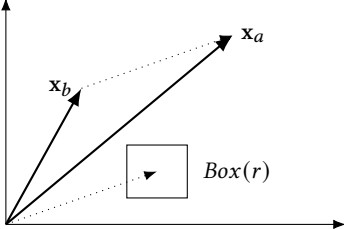

**Figure 1: Illustration of embedding roles as boxes**

According to our definition, $\mathcal{EL}^{++}$-closed embeddings can be extended to encompass the embedding of any $\mathcal{EL}^{++}$-concept and, consequently, can be applied to complex tasks such as axiom learning and query answering. However, as we demonstrated, the current state-of-the-art method, Box$^2$EL, is not $\mathcal{EL}^{++}$-closed. While simpler methods like ELBE and BoxEL achieve $\mathcal{EL}^{++}$-closure, they fail to represent complex many-to-many relationships and their performance is limited as shown in the experiments over real-world ontologies. In the following section, we present *TransBox*, our proposed $\mathcal{EL}^{++}$-closed method aimed at addressing these challenges.

## 4 Method: TransBox

We now introduce *TransBox*, our method for constructing geometric models of a given $\mathcal{EL}^{++}$ ontology $O$. This section is structured as follows. First, we present the basic framework of TransBox in Section 4.1. In Section 4.2, we demonstrate that TransBox is $\mathcal{EL}^{++}$-closed. Sections 4.3 and 4.4 introduce two enhancements that improve the learned embeddings and handling of box intersections. The training procedure for TransBox is outlined in Section 4.5.

### 4.1 Geometric Construction

*Concept and individual.* In TransBox, we embed each atomic concept $A \in \mathsf{N_C}$ to a box $Box(A)$, which is defined as an axis-aligned hyperrectangle in the n-dimensional linear space $\mathbb{R}^n$. As in previous works [17, 26, 31], each box $Box(A)$ is represented by a center $\mathbf{c}(A) = (c_1, \ldots, c_n) \in \mathbb{R}^n$ and offset $\mathbf{o}(A) = (o_1, \ldots, o_n) \in \mathbb{R}^n_{\geq 0}$. Formally, $Box(A) \subseteq \mathbb{R}^n$ is the area defined by:

$$Box(A) = \{\mathbf{x} \in \mathbb{R}^n \mid \mathbf{c}(A) - \mathbf{o}(A) \leq \mathbf{x} \leq \mathbf{c}(A) + \mathbf{o}(A)\}, \quad (5)$$

where $\leq$ is the element-wise comparison. Each individual $a \in \mathsf{N_I}$ is embedded as a points $\mathbf{x}_a \in \mathbb{R}^n$.

*Role.* Each role $r \in \mathsf{N_R}$ is also associated with a box $Box(r) \subseteq \mathbb{R}^n$, with the semantics defined as follows: For each pair of embedded instances $\mathbf{x}_a, \mathbf{x}_b \in \mathbb{R}^n$, we have $r(a, b)$ is true if $\mathbf{x}_a - \mathbf{x}_b \in Box(r)$ (see Figure 1 for an illustration). Formally, we embed each $r$ to an area in $E_{Box(r)} \subseteq \mathbb{R}^n \times \mathbb{R}^n$ defined by:

$$E_{Box(r)} = \{(\mathbf{x}, \mathbf{y}) \mid \mathbf{x}, \mathbf{y} \in \mathbb{R}^n, \mathbf{x} - \mathbf{y} \in Box(r)\}. \quad (6)$$

*Expressiveness.* TransBox extends models based on TransE [7] by using a set of translations defined by the vectors in $Box(r)$. This extension allows for more flexible modeling of one-to-many, many-to-one, and many-to-many roles, which cannot be handled by TransE-based models like ELEM and ELBE. Additionally, our framework captures role embeddings more effectively: (1) it preserves

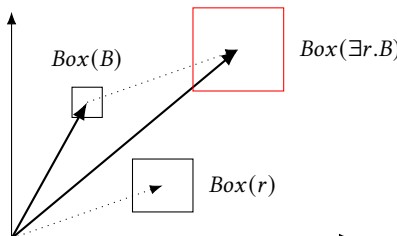

**Figure 2: Illustration of** $Box(\exists r.B)$

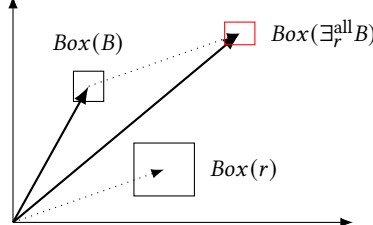

**Figure 3: Illustration of** $Box(\exists_r^{\mathbf{all}} B)$

the ability to represent role compositions, as we will demonstrate in Section 4.2; (2) it naturally expresses role inclusion $r \sqsubseteq s$ as $Box(r) \subseteq Box(s)$, while allowing for different embeddings of $r$ and $s$. This can not be supported by previous methods except for Box²EL.

*Model Complexity.* In an $n$-dimensional space, TransBox requires $2n(N_C + N_R)$ parameters to represent the bounding boxes for atomic concepts and roles, along with additional $nN_I$ parameters to encode individuals. Here, $N_C$, $N_R$, and $N_I$ refer to the number of concepts, roles, and individuals in the ontology $O$, respectively. Thus, the total model complexity of TransBox is $O(n(2N_C + 2N_R + N_I))$.

## 4.2 $\mathcal{EL}^{++}$-closeness

In the TransBox framework, the candidate space for representing concepts $\mathcal{E}_C{}^3$ comprises all boxes in $\mathbb{R}^n$, and the candidate space for representing roles $\mathcal{E}_R$ is defined as a subset $E_{Box(r)} \subseteq \mathbb{R}^n \times \mathbb{R}^n$, as specified in Equation (6). To demonstrate that TransBox is $\mathcal{EL}^{++}$-closed, it suffices to verify the three conditions outlined in Definition 3.1 for any boxes $Box(A)$, $Box(B) \in \mathcal{E}_C$ and subsets $E_{Box(r)}$, $E_{Box(t)} \in \mathcal{E}_R$ as follows.

(1) *Closed under conjunction:* $Box(A) \cap Box(B)$ is always a box, and thus belongs to $\mathcal{E}_C$. Therefore, the first condition of Definition 3.1 holds. We denote by $Box(A \sqcap B) := Box(A) \cap Box(B)$;

(2) *Closed under existential qualification:* The second condition of Definition 3.1 holds according to the following proposition:

> PROPOSITION 4.1. *Let* $S = Box(B)$ *and* $E = E_{Box(r)}$*, and let*
>
> $$\exists_E S = \{\mathbf{x} \mid \exists \mathbf{y} \in \mathbb{R}^n : (\mathbf{x}, \mathbf{y}) \in E\}.$$
>
> *Then* $\exists_E S$ *is a box with center* $\mathbf{c}(r) + \mathbf{c}(B)$ *and offset* $\mathbf{o}(r) + \mathbf{o}(B)$*, and thus, we have* $\exists_E S \in \mathcal{E}_C$.

In the remainder of this paper, we will denote $Box(\exists r.B) := \exists_{E_{Box(r)}} Box(B)$, given that $r$ and $B$ are embedded as $Box(r)$ and $Box(B)$, respectively. For an illustration, see Figure 2.

(3) *Closed under role composition:* The third condition of Definition 3.1 holds based on the following proposition:

> PROPOSITION 4.2. *Let* $Box(r \circ t)$ *be the box with center* $\mathbf{c}(r) + \mathbf{c}(t)$ *and offset* $\mathbf{o}(r) + \mathbf{o}(t)$*. Then, we have* $E_{Box(r \circ t)} = E_{Box(r)} \circ E_{Box(t)}$*, where the composition is defined as* $E_{Box(r)} \circ E_{Box(t)} = \{(\mathbf{x}, \mathbf{z}) \mid \exists \mathbf{y} \in \mathbb{R}^n : (\mathbf{x}, \mathbf{y}) \in Box(r) \text{ and } (\mathbf{y}, \mathbf{z}) \in Box(t)\}$*. Thus,* $E_{Box(r)} \circ E_{Box(t)} \in \mathcal{E}_R$.

## 4.3 Semantic Enhancement

*Semantic enhancement* refers to replacing the ontology $O$ with a "stronger" version, $O^{\text{stg}}$, during the training process. We consider $O^{\text{stg}}$ stronger than $O$ if it derives all axioms in $O$ (*i.e.,* $O^{\text{stg}} \models \alpha$ for any $\alpha \in O$). This enhancement can lead to better results by imposing stronger constraints (an example is provided in Section 6.1).

To realize the semantic enhancements, we introduce a new logical operator, $\exists^{\text{all}}$, which denotes individuals related by a role to every individual in a concept [4]. Therefore, $Box(\exists_r^{\text{all}} B)$, as illustrated in Figure 3, is formally defined by:

$$Box(\exists_r^{\text{all}} B) = \{\mathbf{x} \in \mathbb{R}^n \mid \forall \mathbf{y} \in Box(B) : \mathbf{x} - \mathbf{y} \in Box(r)\}.$$

> PROPOSITION 4.3. $Box(\exists_r^{all} B)$ *is a box with center* $\mathbf{c}(r) + \mathbf{c}(B)$ *and offset* $\max\{\mathbf{0}, \mathbf{o}(r) - \mathbf{o}(B)\}$*, and we have* $Box(\exists_r^{all} B) \subseteq Box(\exists r.B)$.

Based on Proposition 4.3, we perform a semantic enhancement on a given ontology $O$ by replacing any axiom $\alpha \in O$ of the form $C \sqsubseteq D$ with $C \sqsubseteq D^{\text{stg}}$, where $D^{\text{stg}}$ is obtained by substituting each occurrence of $\exists r$ with $\exists_r^{\text{all}}$. Such an operation is guarantee to be an enhancement by the following result:

> PROPOSITION 4.4. *Let* $O^{stg}$ *be the collection of all axioms* $C \sqsubseteq D^{stg}$ *obtained as described above. Then, for any* $\alpha \in O$*, we have* $O^{stg} \models \alpha$.

It is important to note that only TransBox can apply the above enhancement, as it embeds roles as a set of translations represented by boxes. In contrast, methods like ELBE and BoxEL would collapse into TransE with this enhancement. This occurs because ELBE and BoxEL embed roles as a single translation, meaning $Box(r)$ is reduced to a single point, which would similarly reduce $Box(A)$ to a single point under the same enhancement.

## 4.4 Enhancing Box Intersections

The standard intersection of boxes tends to become empty as the space grows exponentially sparser with increasing dimensions. Consequently, the likelihood of finding intersecting boxes decreases exponentially, as demonstrated in Theorem 4.5. However, the existing solutions to such a problem violate the objective of finding valid geometric models. For example, in implementation, Box²EL defines the intersection of two disjoint intervals, such as $[-1, 0]$ and $[2, 3]$, as $[0, 2]$, disregarding the fact that they do not overlap. BoxEL addresses this issue by defining specific volumes for such cases; however, it only works for concepts of the form $A \sqcap B$ and cannot be extended to handle more complex cases, such as $A \sqcap B \sqcap B'$.

---

³All singletons $\{\mathbf{x}\} \in \mathcal{E}_C$ by setting the offset to $\mathbf{0}$. We also assume that $\emptyset, \mathbb{R}^n \in \mathcal{E}_C$.

⁴$\exists_r^{\text{all}}$ is different from $\forall r$, see Appendix C for an example.

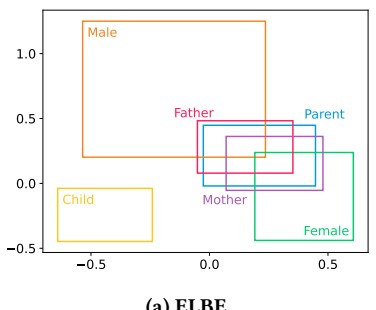

(a) ELBE

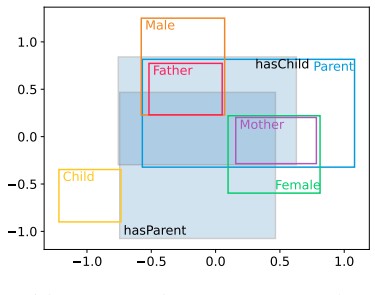

(b) TransBox (w/o Enhancement)

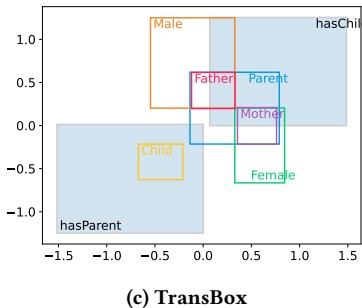

(c) TransBox

**Figure 4: Embedding visualization of ELBE and TransBox for the family ontology in the 2-dimensional case. The box embeddings for roles in TransBox are highlighted in blue. 'w/o Enhancement' refers to training TransBox without the semantic and intersection enhancements.**

$$Father \sqsubseteq Male \sqcap Parent$$
$$Male \sqcap Parent \sqsubseteq Father$$
$$Male \sqcap Female \sqsubseteq \bot$$
$$Child \sqsubseteq \exists hasParent.Mother$$
$$Parent \sqsubseteq \exists hasChild.Child$$

$$Mother \sqsubseteq Female \sqcap Parent$$
$$Female \sqcap Parent \sqsubseteq Mother$$
$$Parent \sqcap Child \sqsubseteq \bot$$
$$Child \sqsubseteq \exists hasParent.Father$$

**Figure 5: Family Ontology**

THEOREM 4.5. *Let Box and Box′ be two randomly generated boxes in an n-dimensional space, with centers uniformly selected from $[-1, 1]^n$ and offsets chosen from $(0, 1]^n$. Then, we have the possiblity $P(Box \cap Box' \neq \emptyset) = (2/3)^n$. Thus, for $n \geq 50$, we have $P(Box \cap Box') < 1.6 \times 10^{-9}$.*

To address this issue, we propose extending the box representation from $\mathbb{R}^n$ to $(\mathbb{R} \cup \{\emptyset\})^n$. This solution is more natural compared to existing approaches and better aligns with the goal of developing geometric models.

*Extended Box intersections over $(\mathbb{R} \cup \{\emptyset\})^n$.* First, noted that a box $Box \subseteq \mathbb{R}^n$ can be regarded as a Cartesian product of intervals:

$$Box = I_1 \times I_2 \times \ldots \times I_n, \tag{7}$$

where $I_i := [c_i - o_i, c_i + o_i]$ $(1 \leq i \leq n)$ is a interval with $c_i, o_i$ the coordinates of center $\mathbf{c}$ and offset $\mathbf{o}$ the $Box$.

We define $I^* := I \cup \{\emptyset\}$, and the boxes over $(\mathbb{R} \cup \{\emptyset\})^n$ as:

$$Box = I_1^* \times I_2^* \times \ldots \times I_n^* \tag{8}$$

Then, the intersection of two boxes $Box, Box' \subseteq (\mathbb{R} \cup \{\emptyset\})^n$ is defined by taking the intersection over each component. Formally:

$$Box \cap Box' = (I_1^* \cap I_1^{*'}) \times \ldots \times (I_n^* \cap I_n^{*'}). \tag{9}$$

Under such a setting, we say $Box$ and $Box'$ is disjoint iff $Box \cap Box' = \{\emptyset\} \times \ldots \times \{\emptyset\}$. One can see that, by our extension, we could have non-empty intersection of $Box$ and $Box'$ even if $I_i^* \cap I_i^{*'} = \emptyset$ for several $1 \leq i \leq n$.

## 4.5 Training

*Distance and inclusion loss of Boxes.* As shown in [22, 26], the distance of two boxes can be described by the following equation:

$$d(Box, Box') = |\mathbf{c}_1 - \mathbf{c}_2| - \mathbf{o}_1 - \mathbf{o}_2. \tag{10}$$

Note that $d(Box, Box') \in \mathbb{R}^n$ is a vector that captures the minimal difference between two points between $Box$ and $Box'$, and a negative $d(Box, Box')$ indicates that $Box$ and $Box'$ have non-empty intersection.

The inclusion loss that determines whether one box is included in another is defined as follows. We assign each box on $(\mathbb{R} \cup \{\emptyset\})^n$ a binary mask $\mathbf{m} = (m_1, \ldots, m_n)$, where $m_j = 1$ if $I_j^* \neq \emptyset$, and $m_j = 0$ otherwise. Using this mask, we extend the standard inclusion loss to boxes in $(\mathbb{R} \cup \{\emptyset\})^n$ as:

$$\mathcal{L}_{\subseteq}(Box, Box') := ||\mathbf{o}(Box) * \mathbf{m} * (1 - \mathbf{m}')|| + \\ || \max\{\mathbf{0}, (d(Box, Box') + 2\mathbf{o}(Box)) * \mathbf{m} * \mathbf{m}' - \gamma\}|| \tag{11}$$

where (1) The first term encourages the offset of $Box$ to shrink to zero in dimensions where $Box'$ is empty (i.e., where $m_j' = 0$); (2) The second term represents the standard inclusion loss [17], but is restricted to the dimensions where both $Box$ and $Box'$ are non-empty (i.e., $m_j m_j' = 1$ in those dimensions). Note that when the mask vectors have a value of 1 in all dimensions, the above inclusion loss becomes equivalent to the standard one over $\mathbb{R}^n$.

*Axiom Loss.* The model is trained by the following loss function for two kinds of axioms with semantic enhancement applied:

(1) General concept inclusion axioms $C \sqsubseteq D$: We define the loss as the inclusion loss between $Box(C)$ and $Box(D)$:

$$\mathcal{L}(C \sqsubseteq D) = \mathcal{L}_{\subseteq}(Box(C), Box(D)). \tag{12}$$

(2) Role inclusion axioms $r_1 \circ \ldots \circ r_n \sqsubseteq t$ $(n = 1, 2)$:

$$\mathcal{L}(r_1 \circ \ldots \circ r_n \sqsubseteq t) = \mathcal{L}_{\subseteq}(Box(r_1 \circ \ldots \circ r_n), Box(t)). \tag{13}$$

*Negative Sampling.* Similar to Knowledge Graph Embedding methods (e.g., [7]) and existing works on Ontology Embedding (e.g., [22]), we use negative sampling to avoid trivial embeddings and enhance the learned embeddings' quality. The negative samples are built by replacing some atomic concepts in an axiom randomly by a different concepts. Specifically, following [17], we generate native samples of the form $A' \sqsubseteq \exists r.B'$ by replacing $A$ with $A'$ and $B$ with $B'$ for each axiom $A \sqsubseteq \exists r.B \in O$. Moreover, a loss has been introduced to discourage $A' \sqsubseteq \exists r.B'$ from holding, as $Box(A') \nsubseteq Box(\exists r.B')$, using the following distance function that makes the minimal distance between $Box(A')$ and $Box(\exists r.B')$ close

**Table 1: Distribution of generated complex axioms**

| Dataset | $A \sqsubseteq D$ | $C \sqsubseteq A$ | $C \sqsubseteq D$ |
|---|---|---|---|
| GALEN | 200 | 630 | 166 |
| GO | 397 | 347 | 256 |
| ANATOMY | 897 | 32 | 69 |

to 1 in each coordinate:

$$\mathcal{L}_{\not\sqsubseteq}(A' \sqsubseteq \exists r.B') = (1 - \| \max\{\mathbf{0}, -d(Box(A'), Box(\exists r.B')) - \gamma\}\|)^2.$$

*Regularization.* Similar to TransE and ELEM, we introduce a regularization term that encourages the norm of the centers of concept boxes to be close to 1. That is:

$$\mathcal{L}_{con}^{reg}(\alpha) = \sum_{A \text{ appears in } \alpha} \|\mathbf{c}(A) - \mathbf{1}\|. \tag{14}$$

In Conclusion, the final loss function is defined as:

$$\mathcal{L} = \frac{1}{N}\left( \sum_{\alpha \in O} (\mathcal{L}(\alpha) + \lambda \cdot \mathcal{L}_{con}^{reg}(\alpha))^2 + \sum_{\text{neg sample } \beta} \mathcal{L}_{\not\sqsubseteq}(\beta) \right). \tag{15}$$

where $\lambda \geq 0$ is a regularization factor, $N$ the batch size.

## 5 Soundness

Let $O$ be an $\mathcal{EL}^{++}$ ontology. We can extend any TransBox embedding of $O$ to an interpretation $\mathcal{I}_g$ following the $\mathcal{EL}^{++}$ semantics introduced in Section 2.1, where the atomic concepts in $O$ are represented by $Box(A)$, and the roles in $O$ by $E_{Box(r)}$, as specified in Equation (6).

By the following result, we show that Transbox *is sound* in the sense that such an interpretation $\mathcal{I}_g$ is a geometric model of $O$ when the loss for any axiom in $O$ is zero.

THEOREM 5.1 (SOUNDNESS). *If for every axiom $\alpha \in O$, the loss defined by Transbox is 0 (i.e., $\mathcal{L}(\alpha) = 0$), then $\mathcal{I}_g$ is a geometric model of $O$.*

PROOF. Since Transbox is closed under $\mathcal{EL}^{++}$, for any (complex) $\mathcal{EL}^{++}$-concept $C$, we have $Box(C) = C^{\mathcal{I}_g}$ by definition. Now, consider any axiom $\alpha$ of the form $C \sqsubseteq D$. By our definition, $\mathcal{L}(C \sqsubseteq D) = 0$ if and only if $Box(C) \subseteq Box(D)$. This implies that $C^{\mathcal{I}_g} = Box(C) \subseteq Box(D) = D^{\mathcal{I}_g}$. Similarly, for role composition axioms of the form $r_1 \circ r_2 \sqsubseteq t$, we have $(r_1 \circ r_2)^{\mathcal{I}_g} \subseteq t^{\mathcal{I}_g}$ whenever $\mathcal{L}(r_1 \circ r_2 \sqsubseteq t) = 0$.

Thus, we conclude that $\mathcal{I}_g$ is indeed a geometric model of $O$. □

The theorem is not affected by either semantic enhancement or extended intersections. For the semantic enhancement, this is because $\mathcal{L}(A \sqsubseteq \exists_r^{all}.B) = 0$ implies $\mathcal{L}(A \sqsubseteq \exists r.B) = 0$. For the extended intersections, the theorem remains as $\mathcal{L}(Box, Box') = 0$ continues to imply that $Box \subseteq Box'$ for boxes over $(\mathbb{R} \cup \{\emptyset\})^n$.

## 6 Evaluation Results

### 6.1 Proof of Concept: Family Ontology

To evaluate and demonstrate the expressiveness of the TransBox embedding method, we use a simple family ontology and compare the embedding results with ELBE [26]. For better illustration, we

**Table 2: Overall comparison of prediction of complex axioms**

| | Model | H@1 | H@10 | H@100 | Med | MRR | MR | AUC |
|---|---|---|---|---|---|---|---|---|
| GALEN | BoxEL | 0.00 | 0.01 | 0.05 | 959 | 0.00 | 4794 | 0.54 |
| | ELBE | 0.00 | 0.00 | 0.00 | 995 | 0.00 | 5054 | 0.51 |
| | TransBox | **0.01** | **0.05** | **0.15** | **727** | **0.02** | **2769** | **0.73** |
| GO | BoxEL | 0.00 | 0.01 | 0.05 | 982 | 0.00 | 4516 | 0.67 |
| | ELBE | 0.05 | 0.09 | 0.15 | 1035 | 0.07 | 5217 | 0.61 |
| | TransBox | **0.16** | **0.41** | **0.65** | **30** | **0.25** | **717** | **0.95** |
| Anatomy | BoxEL | 0.00 | 0.00 | 0.05 | 1020 | 0.00 | 19744 | 0.60 |
| | ELBE | 0.05 | 0.08 | 0.10 | 995 | 0.06 | 15661 | 0.68 |
| | TransBox | **0.26** | **0.55** | **0.69** | **7** | **0.35** | **622** | **0.99** |

train all embeddings in a 2-dimensional space, and add visualization loss below into the final loss functions to avoid overly small boxes as in Box$^2$EL [17].

$$\mathcal{L}_V = \frac{1}{n|\mathsf{N}_C|} \sum_{A \in \mathsf{N}_C} \sum_{i=1}^{n} \max\{0, 0.2 - o(Box(A))_i\}.$$

In this simple experiment, we set the margin to $\gamma = 0$, regularization factor $\lambda = 0$, and omit negative sampling. The resulting embeddings are shown in Figure 4.

As shown in Figure 4a, the ELBE embeddings fail to capture the disjointness between *Father* and *Mother*. This is because, in ELBE, roles are embedded as translations defined by a single vector. Consequently, the embedded boxes of *Father* and *Mother* must be close to the translated box $Box(Child) + \mathbf{v}_{hasParent}$ due to the axioms $Child \sqsubseteq \exists hasParent.Mother$ and $Child \sqsubseteq \exists hasParent.Father$.

In contrast, TransBox embeddings (Figures 4b and 4c) resolve this issue by utilizing multi-transition representations within $Box(r)$. This approach allows TransBox to correctly learn distinct embeddings for each concept. Additionally, applying the semantic enhancements described in Section 4.3 further improves the embeddings of roles. Specifically, in Figure 4c, we observe that the embedding $Box(hasParent) \approx -Box(hasChild)$ well captures the fact that the role *hasParent* is the inverse of *hasChild*.

### 6.2 Axiom Prediction

*Benchmark.* Following with prior research [17, 31], we utilize three normalized biomedical ontologies: GALEN [27], Gene Ontology (GO) [1], and Anatomy (Uberon) [25] in our study. Training, validation, and testing employ the established 80/10/10 partition as in [17] for predicting normalized axioms. For predicting complex axioms, however, we adopt a distinct test set generated via *forgetting* [21]. The forgetting tool LETHE [20] is chosen as its results are typically more compact and readable compared to others [8, 32]. The statistics of generated axioms are shown in Table 1. More details of the test set generation can be found in Appendix D.

*Baselines.* We primarily compare TransBox with leading box-based methods: BoxEL [31], ELBE [26], and Box$^2$EL [17]. These approaches are closely aligned with our objectives and provide a strong basis for evaluating the effectiveness of TransBox. Ball-based methods like ELEM [22] and ELEM++ [24], while relevant, are only applicable to specific axioms, and thus comparisons with them are discussed in Appendix G.

**Table 3: Comparasion over subtask: $* \sqsubseteq ?A$**

| | Model | H@1 | H@10 | H@100 | Med | MRR | MR | AUC |
|---|---|---|---|---|---|---|---|---|
| GALEN | BoxEL | 0.00 | 0.00 | 0.00 | 12600 | 0.00 | 11714 | 0.49 |
| | ELBE | 0.00 | 0.00 | 0.00 | 16779 | 0.00 | 11904 | 0.50 |
| | TransBox | 0.00 | **0.07** | **0.15** | **4887** | **0.02** | **6683** | **0.71** |
| GO | BoxEL | 0.00 | 0.00 | 0.00 | 9439 | 0.00 | 11937 | 0.74 |
| | ELBE | 0.00 | 0.00 | 0.00 | 16230 | 0.00 | 17480 | 0.62 |
| | TransBox | **0.02** | **0.65** | **0.86** | **4** | **0.24** | **1216** | **0.97** |
| Anatomy | BoxEL | 0.00 | 0.00 | 0.09 | 1299 | 0.00 | 6358 | 0.95 |
| | ELBE | 0.00 | 0.05 | 0.08 | 24293 | 0.02 | 29700 | 0.73 |
| | TransBox | **0.03** | **0.43** | **0.76** | **16** | **0.16** | **970** | **0.99** |

**Table 4: Comparasion over subtask: $?A \sqsubseteq *$**

| | Model | H@1 | H@10 | H@100 | Med | MRR | MR | AUC |
|---|---|---|---|---|---|---|---|---|
| GALEN | BoxEL | 0.00 | 0.00 | 0.00 | 6176 | 0.00 | 7426 | 0.68 |
| | ELBE | 0.00 | 0.00 | 0.00 | 7775 | 0.00 | 8931 | 0.62 |
| | TransBox | **0.07** | **0.16** | **0.35** | **558** | **0.10** | **3164** | **0.87** |
| GO | BoxEL | 0.00 | 0.00 | 0.01 | 17396 | 0.00 | 19249 | 0.58 |
| | ELBE | 0.00 | 0.00 | 0.00 | 16162 | 0.00 | 16836 | 0.64 |
| | TransBox | 0.00 | **0.12** | **0.56** | **61** | **0.04** | **2566** | **0.95** |
| Anatomy | BoxEL | 0.00 | 0.00 | 0.00 | 41038 | 0.00 | 42455 | 0.60 |
| | ELBE | 0.00 | 0.00 | 0.00 | 27471 | 0.00 | 32689 | 0.69 |
| | TransBox | **0.17** | **0.70** | **0.90** | **5** | **0.33** | **876** | **0.99** |

**Table 5: Comparasion over subtask: $* \sqsubseteq ?C$**

| | Model | H@1 | H@10 | H@100 | Med | MRR | MR | AUC |
|---|---|---|---|---|---|---|---|---|
| GALEN | BoxEL | 0.00 | 0.03 | 0.27 | 184 | 0.02 | 368 | 0.68 |
| | ELBE | 0.00 | 0.00 | 0.01 | 334 | 0.00 | 459 | 0.61 |
| | TransBox | **0.01** | **0.06** | **0.34** | **204** | **0.04** | **200** | **0.83** |
| GO | BoxEL | 0.00 | 0.02 | 0.15 | 561 | 0.01 | 611 | 0.58 |
| | ELBE | 0.02 | 0.05 | 0.16 | 664 | 0.03 | 593 | 0.59 |
| | TransBox | **0.25** | **0.52** | **0.62** | **8** | **0.34** | **388** | **0.73** |
| Anatomy | BoxEL | 0.00 | 0.01 | 0.10 | 497 | 0.01 | 503 | 0.53 |
| | ELBE | 0.09 | 0.12 | 0.15 | 564 | 0.11 | 538 | 0.50 |
| | TransBox | **0.37** | **0.46** | **0.48** | **365** | **0.41** | **406** | **0.62** |

**Table 6: Comparasion over subtask: $?C \sqsubseteq *$**

| | Model | H@1 | H@10 | H@100 | Med | MRR | MR | AUC |
|---|---|---|---|---|---|---|---|---|
| GALEN | BoxEL | 0.00 | 0.00 | 0.00 | **684** | 0.00 | **680** | **0.42** |
| | ELBE | 0.00 | 0.00 | 0.00 | 763 | 0.00 | 761 | 0.35 |
| | TransBox | 0.00 | 0.00 | 0.00 | 758 | 0.00 | 747 | 0.36 |
| GO | BoxEL | 0.00 | 0.00 | 0.00 | 823 | 0.00 | 827 | 0.43 |
| | ELBE | 0.11 | 0.18 | 0.25 | 1046 | 0.14 | 777 | 0.47 |
| | TransBox | **0.19** | **0.29** | **0.60** | **68** | **0.23** | **311** | **0.79** |
| Anatomy | BoxEL | 0.00 | 0.00 | 0.00 | 974 | 0.00 | 974 | 0.09 |
| | ELBE | **0.18** | **0.38** | 0.57 | 56 | **0.25** | 425 | 0.61 |
| | TransBox | 0.09 | 0.21 | **0.69** | **45** | 0.13 | **254** | **0.77** |

*Evaluation Metric.* In line with previous works [17, 22, 26, 31], we assess the performance of ontology embeddings using a variety of ranking-based metrics on the test set. Following [17], we rank the candidates based on a score function defined as the negative value of the distance between the embeddings of the concepts on both sides of the subsumption (detailed in Appendix F). A higher score indicates a more likely axiom. To evaluate the performance of different methods, we record the rank of the correct answer and report it using several standard evaluation metrics: Hits@k (H@k) for $k \in \{1, 10, 100\}$, median rank (Med), mean reciprocal rank (MRR), mean rank (MR), and area under the ROC curve (AUC).

*Experimental Protocol.* We train all models with dimensions $d \in \{25, 50, 100, 200\}$, margins $\gamma \in \{0, 0.05, 0.1, 0.15\}$, learning rates $l_r \in \{0.0005, 0.005, 0.01\}$, and regularization factor $\lambda = 1$, using the Adam optimizer [19] for 5,000 epochs. All reported results represent the average performance across 10 random runs, with the optimum hyper parameters selected based on validation set performance. Our experiments are based on a re-implementation of [17], with a corrected evaluation on axioms of the form $A \sqcap B \sqsubseteq B'$. More details can be found in Appendix E.

#### 6.2.1 Prediction of complex axioms.

*Evaluation Task.* As shown in Table 1, the test set for the complex axiom prediction task consists of three types of axioms: $A \sqsubseteq C$, $C \sqsubseteq A$, and $C \sqsubseteq D$, where $A$ represents an atomic concept, and $C$ and $D$ are complex $\mathcal{EL}^{++}$-concepts. Based on this structure, we define four different evaluation tasks: $* \sqsubseteq ?A$, $?A \sqsubseteq *$, $* \sqsubseteq ?C$, $?C \sqsubseteq *$, where $?A$ (resp. $?C$) refers to a query for the atomic concept $A$

(resp. the complex concept $C$) on one side of the subsumption, and $*$ represents the candidate concepts on the other side. For tasks involving $A$, the candidate set consists of all atomic concepts in the ontology. For tasks involving $C$, the candidates include all complex concepts $C$ or $D$ present in the test set.

*Results.* The overall result is summarized in Table 2, and the result of each subtask is presented in Tables 3, 4, 5 and 6. Note that for the task of predicting complex axioms, we can only make a comparison with $\mathcal{EL}^{++}$-closed methods BoxEL and ELBE.

We observe that TransBox outperforms all the existing $\mathcal{EL}^{++}$-closed methods, with particularly notable improvements on the GO and Anatomy datasets. For instance, the median rank (Med) improves from over 900 to below 30, while the mean rank (MR) decreases by more than 80% in GO and 95% in Anatomy. A detailed analysis of each subtask, shown in Tables 4 to 6, highlights consistent performance gains across nearly all the cases. This is especially evident in the prediction of atomic concepts (i.e., $* \sqsubseteq ?A$, $?A \sqsubseteq *$), where median ranks drop from thousands or tens of thousands to under 100 or even below 10. As expected, predicting atomic concepts performs better than predicting complex concepts. This is because as concept complexity increases, the embeddings' ability to capture meaning diminishes due to accumulated errors in the composition process.

#### 6.2.2 Normalized Axioms Prediction.
The overall results for the four types of normalized axioms ($A \sqsubseteq B$, $A \sqsubseteq B \sqsubseteq B'$, $A \sqsubseteq \exists r.B$, and $\exists r.B \sqsubseteq A$) are presented in Table 7. While TransBox achieved the best overall performance among $\mathcal{EL}^{++}$ methods, it is less competitive compared to non-$\mathcal{EL}^{++}$ methods for predicting normalized

**Table 7: Overall comparison of $\mathcal{EL}$-closed method on normalized axioms**

| | Model | H@1 | H@10 | H@100 | Med | MRR | MR | AUC |
|---|---|---|---|---|---|---|---|---|
| GALEN | Box²EL | **0.04** | **0.17** | **0.31** | 1360 | **0.08** | 5183 | 0.78 |
| | BoxEL | 0.00 | 0.03 | 0.16 | 4750 | 0.01 | 7213 | 0.69 |
| | ELBE | 0.01 | 0.07 | 0.14 | 5340 | 0.03 | 7447 | 0.68 |
| | TransBox | 0.01 | 0.08 | 0.18 | 4986 | 0.03 | 7227 | 0.69 |
| GO | Box²EL | 0.02 | **0.17** | **0.52** | **86** | **0.07** | 4593 | **0.90** |
| | BoxEL | 0.01 | 0.06 | 0.08 | 8572 | 0.03 | 15116 | 0.67 |
| | ELBE | **0.03** | 0.13 | 0.19 | 6836 | 0.06 | 10809 | 0.76 |
| | TransBox | 0.01 | 0.16 | 0.35 | 1503 | 0.06 | 8199 | 0.82 |
| Anatomy | Box²EL | **0.07** | **0.34** | **0.65** | 27 | **0.15** | 2918 | **0.97** |
| | BoxEL | 0.03 | 0.11 | 0.25 | 1527 | 0.06 | 11930 | 0.89 |
| | ELBE | 0.02 | 0.27 | 0.47 | 162 | 0.10 | 9562 | 0.91 |
| | TransBox | 0.03 | 0.35 | 0.62 | 29 | 0.13 | 8186 | 0.92 |

**Table 8: Comparasion over prediction:** $A \sqcap B \sqsubseteq ?B'$

| | Model | H@1 | H@10 | H@100 | Med | MRR | MR | AUC |
|---|---|---|---|---|---|---|---|---|
| GALEN | Box²EL | 0.00 | 0.00 | 0.00 | 10910 | 0.00 | 11063 | 0.52 |
| | BoxEL | 0.00 | 0.00 | 0.01 | 11217 | 0.00 | 11437 | 0.51 |
| | ELBE | 0.00 | 0.00 | 0.00 | 10898 | 0.00 | 11053 | 0.52 |
| | TransBox | **0.03** | **0.21** | **0.39** | **996** | **0.08** | **5112** | **0.78** |
| GO | Box²EL | 0.00 | 0.00 | 0.01 | 14273 | 0.00 | 15566 | 0.66 |
| | BoxEL | 0.00 | 0.00 | 0.00 | 22263 | 0.00 | 22569 | 0.51 |
| | ELBE | 0.00 | 0.00 | 0.01 | 14712 | 0.00 | 16109 | 0.65 |
| | TransBox | **0.06** | **0.60** | **0.77** | **7** | **0.21** | **2783** | **0.94** |
| Anatomy | Box²EL | 0.00 | 0.01 | 0.02 | 29327 | 0.00 | 34052 | 0.68 |
| | BoxEL | 0.00 | 0.00 | 0.01 | 16982 | 0.00 | 24917 | 0.77 |
| | ELBE | 0.01 | 0.02 | 0.06 | 22228 | 0.01 | 28524 | 0.73 |
| | TransBox | **0.05** | **0.29** | **0.57** | **56** | **0.12** | **2215** | **0.98** |

axioms. This is expected, as non-$\mathcal{EL}^{++}$ methods, unconstrained by extensional capabilities, are often better suited to capture normalized axioms due to their specific design. For instance, Box²EL's use of bump vectors provides enhanced capacity for embedding concepts and roles to fit axioms. However, this approach is restricted to atomic concepts for which bump vectors are explicitly defined. Furthermore, it introduces greater model complexity for storing these bump vectors compared to TransBox, with a total complexity of $O(n(3N_C + 4N_R + 2N_I))$ versus $O(n(2N_C + 2N_R + N_I))$.

Notably, thanks to the enhanced box intersection mechanism introduced in Section 4.4, TransBox performs significantly better on predicting $A \sqcap B \sqsubseteq ?B'$, which is an important and common type of axioms that encapsulate the relationships between multiple concepts and their implications. However, since this type of axiom makes up a small portion of our test sets (16.14% in GALEN, 9.33% in GO, and 0.77% in ANATOMY), its contribution to the overall results in Table 7 is underrepresented.

*6.2.3 Ablation Study.* We performed an ablation study to evaluate the effects of semantic enhancement and extended intersection on prediction performance, using the GALEN ontology for all experiments. The results demonstrate that applying intersection enhancement significantly improves performance on tasks involving

**Table 9: Ablation Study:** *SemEn* refers to Semantic Enhancement, and *IntEn* refers to Intersection Enhancement.

| Task | SemEn | IntEn | H@1 | H@10 | H@100 | Med | MRR | MR | AUC |
|---|---|---|---|---|---|---|---|---|---|
| Complex | ✓ | ✓ | **0.01** | **0.05** | **0.15** | **727** | **0.02** | **2769** | **0.73** |
| | | ✓ | 0.01 | 0.04 | 0.12 | 784 | 0.02 | 3043 | 0.71 |
| | ✓ | | 0.00 | 0.00 | 0.01 | 994 | 0.00 | 5061 | 0.51 |
| | | | 0.00 | 0.00 | 0.00 | 994 | 0.00 | 5050 | 0.51 |
| Normalized | ✓ | ✓ | 0.01 | 0.08 | 0.18 | 4986 | 0.03 | 7227 | 0.69 |
| | | ✓ | 0.01 | 0.04 | 0.08 | 9296 | 0.02 | 9931 | 0.57 |
| | ✓ | | 0.01 | **0.09** | **0.20** | 4352 | 0.03 | **6857** | **0.70** |
| | | | 0.01 | 0.05 | 0.10 | 8620 | 0.02 | 9421 | 0.59 |

complex axioms. However, when restricted to normalized axioms, the intersection enhancement shows minimal impact. In such cases, semantic enhancement plays a more critical role. Overall, the combination of both semantic and intersection enhancements yields the best results, highlighting their complementary strengths in improving predictive accuracy.

*6.2.4 Case Study.* In the following query of the form $C \sqsubseteq ?A$ over GALEN, our model correctly identified the correct answer *EndometriosisLesion* with a score of -0.32, ranking 3rd. Note the score is the negative value of the distance. It effectively distinguished *EndometriosisLesion* from other similar conditions such as *EndometrialHypoplasia* (-0.65), *EndometrialRegeneration* (-0.69), and *EndometrialNeoplasia* (-0.64).

EndometrioidStructure ⊓ SnowLeopard

⊓ ∃ *isSpecificConsequenceOf.* MenopauseProcess ⊑?*A*.

This example also underscores the significance of handling complex axioms in practical, real-world applications. In the original, normalized GALEN ontology, *EndometriosisLesion* appears on the right-hand side of two normalized axioms:

Depolarising ⊓ SerumCalciumTest ⊑ EndometriosisLesion

EndometrioidStructure ⊓ LigamentOfUterus ⊑ EndometriosisLesion

While these normalized axioms are informative, the more complex axiom used in the query provides richer context and deeper insights that may not be captured in simpler forms. Additionally, this complexity makes the axiom more intuitive and actionable for medical professionals, aiding in more accurate diagnosis and decision-making.

## 7 Conclusion and Future Work

In this work, we introduced $\mathcal{EL}^{++}$-closed ontology embeddings that can generate embeddings for complex concepts from embeddings of atomic ones, enabling their use in more various tasks. Additionally, we proposed a novel $\mathcal{EL}^{++}$-closed embedding method, *TransBox*, which achieves state-of-the-art performance in predicting complex axioms for three real-world ontologies.

In future work, we plan to extend the closed embeddings to more expressive Description Logics, such as $\mathcal{ALC}$. We are also interested in integrating geometric models with language models, incorporating the textual information of concepts and roles in $\mathcal{EL}^{++}$-closed embeddings for more accurate complex axiom prediction.

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

# A Proofs

## A.1 Proposition 4.1

PROOF. (of Proposition 4.1) By definition, a point $\mathbf{x} \in \exists_E S$ if and only if there exist $\mathbf{y} \in Box(B)$ and $\mathbf{z} \in Box(r)$ such that

$$\mathbf{x} - \mathbf{y} = \mathbf{z}.$$

Since $\mathbf{y} \in Box(B)$, we can express $\mathbf{y}$ as $\mathbf{y} = \mathbf{c}(B) + \mathbf{u}_1$, where $\mathbf{u}_1 \in \mathbb{R}^n$ and $|\mathbf{u}_1| \leq \mathbf{o}(B)$. Here, $|\mathbf{u}_1|$ means the vector obtained by taking the absolute value of each component of $\mathbf{u}_1$. Similarly, $\mathbf{z} \in Box(r)$ can be written as $\mathbf{z} = \mathbf{c}(r) + \mathbf{u}_2$, where $\mathbf{u}_2 \in \mathbb{R}^n$ and $|\mathbf{u}_2| \leq \mathbf{o}(r)$. Substituting these into the equation $\mathbf{x} - \mathbf{y} = \mathbf{z}$, we get:

$$\mathbf{x} - (\mathbf{c}(B) + \mathbf{u}_1) = \mathbf{c}(r) + \mathbf{u}_2.$$

Thus, we have:

$$\mathbf{x} = (\mathbf{c}(B) + \mathbf{c}(r)) + (\mathbf{u}_1 + \mathbf{u}_2).$$

Let $\mathbf{u} = \mathbf{u}_1 + \mathbf{u}_2$, then $\mathbf{u}$ is any point in $\mathbb{R}^n$ such that $|\mathbf{u}| \leq \mathbf{o}(B) + \mathbf{o}(r)$ by construction. Therefore, the set $\exists_E S$ of all possible $\mathbf{x}$ of the above form is exactly a box with center $\mathbf{c}(B) + \mathbf{c}(r)$ and offset $\mathbf{o}(B) + \mathbf{o}(r)$. □

## A.2 Proposition 4.2

PROOF. (of Proposition 4.2) It suffices to show that $(\mathbf{x}, \mathbf{y}) \in Box(r)$ and $(\mathbf{y}, \mathbf{z}) \in Box(t)$ if and only if $\mathbf{x} - \mathbf{z} \in Box(r \circ t)$, which is demonstrated as follows.

Since $\mathbf{x} - \mathbf{y} \in Box(r)$, we can express $\mathbf{x} - \mathbf{y} = \mathbf{c}(r) + \mathbf{u}_1$, where $\mathbf{u}_1 \in \mathbb{R}^n$ and $|\mathbf{u}_1| \leq \mathbf{o}(r)$. Similarly, since $\mathbf{y} - \mathbf{z} \in Box(t)$, we write $\mathbf{y} - \mathbf{z} = \mathbf{c}(t) + \mathbf{u}_2$, where $\mathbf{u}_2 \in \mathbb{R}^n$ and $|\mathbf{u}_2| \leq \mathbf{o}(t)$.

Therefore, we have: $\mathbf{x} - \mathbf{z} = (\mathbf{c}(r) + \mathbf{c}(t)) + (\mathbf{u}_1 + \mathbf{u}_2)$. Let $\mathbf{w} = \mathbf{u}_1 + \mathbf{u}_2$, then $\mathbf{w}$ is a point in $\mathbb{R}^n$ such that $|\mathbf{w}| \leq \mathbf{o}(r) + \mathbf{o}(t)$. By construction, the set of all possible $\mathbf{x} - \mathbf{z}$ is exactly the box $Box(r \circ t)$, with center $\mathbf{c}(r) + \mathbf{c}(t)$ and offset $\mathbf{o}(r) + \mathbf{o}(t)$. □

## A.3 Proposition 4.3

PROOF. (of Proposition 4.3) For any $\mathbf{y} \in Box(B)$, it can be written as $\mathbf{c}(B) + \mathbf{u}_1$, where $|\mathbf{u}_1| \leq \mathbf{o}(B)$. Assume $\mathbf{x} \in Box(\exists_r^{\text{all}} B)$, then $\mathbf{x} - \mathbf{y} = \mathbf{x} - \mathbf{c}(B) - \mathbf{u}_1 \in Box(r)$ for any $|\mathbf{u}_1| \leq \mathbf{o}(B)$. That is:

$$|\mathbf{x} - \mathbf{c}(B) - \mathbf{u}_1 - \mathbf{c}(r)| \leq \mathbf{o}(r), \quad \forall |\mathbf{u}_1| \leq \mathbf{o}(B).$$

Rewriting $\mathbf{x} = \mathbf{c}(B) + \mathbf{c}(r) + \mathbf{u}$, we have $|\mathbf{u} - \mathbf{u}_1| \leq \mathbf{o}(r), \forall |\mathbf{u}_1| \leq \mathbf{o}(B)$. It follows that this holds if and only if $|\mathbf{u}| \leq \max\{\mathbf{0}, \mathbf{o}(r) - \mathbf{o}(B)\}$. In conclusion, $\mathbf{x} \in Box(\exists_r^{\text{all}} B)$ if and only if $\mathbf{x} = \mathbf{c}(B) + \mathbf{c}(r) + \mathbf{u}$ for some $|\mathbf{u}| \leq \max\{\mathbf{0}, \mathbf{o}(r) - \mathbf{o}(B)\}$. This proves the proposition. □

## A.4 Proposition 4.4

PROOF. (of Proposition 4.4) For any interpretation $\mathcal{I}$ of $O$, the following holds:

(1) $A^{\mathcal{I}} \subseteq A^{\mathcal{I}}$ for any atomic concept $A$;
(2) If $D^{\mathcal{I}} \subseteq D^{\mathcal{I}}$, then $(\exists_r^{\text{all}} D)^{\mathcal{I}} \subseteq (\exists r.D)^{\mathcal{I}}$ for any role $r$;
(3) If $D_i^{\mathcal{I}} \subseteq D_i^{\mathcal{I}}$ for $i = 1, 2$, then $D_1^{\mathcal{I}} \cap D_2^{\mathcal{I}} \subseteq D_1^{\mathcal{I}} \cap D_2^{\mathcal{I}}$.

By induction, we conclude that $\models D^{\text{stg}} \sqsubseteq D$. Therefore, for any $C \sqsubseteq D \in O$, we have $\{C \sqsubseteq D^{\text{stg}}\} \models C \sqsubseteq D$, and thus $O' \models C \sqsubseteq D$. □

## A.5 Theorem 4.5

PROOF. (of Theorem 4.5) If we pick two values $x, x'$ (uniform) randomly from $[-1, 1]$, then we have the possibility of $x$ and $x'$ has distance at most $2a$ be:

$$P(|x - x'| \leq 2a) = \frac{4 - (2 - 2a)^2}{4} = 2a - a^2$$

If $Box \cap Box' \neq \emptyset$, assuming $\mathbf{c}, \mathbf{c}'$ are the centers of $Box, Box'$, respectively. Recall that the offset is also choice randomly in $(0, 1]^n$, then we have

$$P(Box \cap Box' \neq \emptyset) = \int_{(a_1, \ldots, a_n) \in (0,1]^n} \prod_{1 \leq i \leq n} P(|\mathbf{c}_i - \mathbf{c}'_i| \leq 2a_i)$$

$$= \int_{(a_1, \ldots, a_n) \in (0,1]^n} \prod_{1 \leq i \leq n} (2a_i - a_i^2)$$

$$= \prod_{1 \leq i \leq n} \left( \int_{0 \leq a_i \leq 1} (2a_i - a_i^2) \right)$$

$$= \prod_{1 \leq i \leq n} [x^2 - \frac{x^3}{3}]_{x=0}^1$$

$$= (\frac{2}{3})^n$$

□

# B Box$^2$EL treat roles as transitive

In Box$^2$EL, the loss of a role-composition axiom is defined by:

$$\mathcal{L}(r_1 \circ r_2 \sqsubseteq t)$$

$$= \frac{\mathcal{L}_{\sqsubseteq}(Head(r_1), Head(t)) + \mathcal{L}_{\sqsubseteq}(Tail(r_2), Tail(t))}{2}$$

$$= 0$$

Therefore, we always have $r \circ r \sqsubseteq r$ holds as the corresponding loss is 0:

$$\mathcal{L}(r \circ r \sqsubseteq r)$$

$$= \frac{\mathcal{L}_{\sqsubseteq}(Head(r), Head(r)) + \mathcal{L}_{\sqsubseteq}(Tail(r), Tail(r))}{2}$$

$$= 0$$

# C Difference between $\exists^{\text{all}}$ and $\forall$

The semantics of the universal quantifier $\forall r$ is defined as:

$$(\forall r.C)^{\mathcal{I}} = \left\{ a \in \Delta^{\mathcal{I}} \mid \forall b \in C^{\mathcal{I}} \text{ such that } (a, b) \in r^{\mathcal{I}}, \ b \in C^{\mathcal{I}} \right\}, \tag{16}$$

where $r$ is a relation and $C$ is a concept.

The difference between $\exists^{\text{all}}$ (exists for all) and $\forall$ (for all) can be illustrated with the following example:

EXAMPLE 4. *Consider the relation passExam and the concepts CS-GraduatedStudent and CSMandatoryCourse. We distinguish between the following three cases:*

- *$\exists_{passExam}$CSMandatoryCourse: someone who has passed at least **one** CS mandatory course.*
- *$\exists_{passExam}^{all}$CSMandatoryCourse: someone who has passed **all** CS mandatory courses.*
- *$\forall_{passExam}$CSMandatoryCourse: someone who has **only** passed CS mandatory courses.*

**Table 10: Sizes of the different ontologies used in our evaluation from [17].**

| Ontology | Classes | Roles | $C \sqsubseteq D$ | $C \sqcap D \sqsubseteq E$ | $C \sqsubseteq \exists r.D$ | $\exists r.C \sqsubseteq D$ | $C \sqcap D \sqsubseteq \bot$ | $r \sqsubseteq s$ | $r_1 \circ r_2 \sqsubseteq s$ |
|---|---|---|---|---|---|---|---|---|---|
| GALEN | 24,353 | 951 | 28,890 | 13,595 | 28,118 | 13,597 | 0 | 958 | 58 |
| GO | 45,907 | 9 | 85,480 | 12,131 | 20,324 | 12,129 | 30 | 3 | 6 |
| Anatomy | 106,495 | 188 | 122,142 | 2,121 | 152,289 | 2,143 | 184 | 89 | 31 |

Then, we have the following axiom:

$$CSGraduatedStudent \sqsubseteq \exists^{all}_{passExam} CSMandatoryCourse,$$

meaning that every CS graduated student must pass all *mandatory courses*. However, the statement

$$CSGraduatedStudent = \forall_{passExam} CSMandatoryCourse$$

does not hold, because a CS graduated student might also pass exams in courses other than the mandatory ones.

## D Generation of complex axiom

*Forgetting* is a non-standard ontology reasoning task that can be regarded as a rewriting process that eliminates unwanted concepts or roles from the ontology while preserving the deductive logical information of the rest.

EXAMPLE 5. *For ontology $O = \{B \sqsubseteq \exists r.B, \; B \sqsubseteq B_1 \sqcap B_2\}$. If we forget the concept $B$, then the result could be $\{A \sqsubseteq \exists r.(B_1 \sqcap B_2)\}$, which can be regarded as a rewriting of $O$ without the forgotten concept $B$, but still preserves the logical information between other concepts and roles $A, B_1, B_2, r$.*

Using forgetting is a robust method to construct our dataset for complex axioms for two main reasons:

(1) The results of forgetting are always valid, as $O \models \alpha$ for any axiom $\alpha$ in some forgetting result of $O$.
(2) Forgetting tends to produce more complex axioms, as shown in the example above.

We use the forgetting tool LETHE [20] to generate our test data, as LETHE's results are typically more compact and readable compared to others [8, 32].

---

**Algorithm 1** Generating axiom test sets

**Require:** Ontology $O$, number of axioms $N = 1000$
**Ensure:** A set of axioms $\mathcal{M}$ with length between 4 and 10
1: $\mathcal{M} \leftarrow \emptyset$ ▷ Initialize the set of selected axioms
2: **while** $|\mathcal{M}| < N$ **do**
3:     $\Sigma \leftarrow$ Randomly select 1,000 atomic concepts from $O$
4:     $O_\Sigma \leftarrow$ Forget$(O, \Sigma)$
5:     **for** each axiom $\alpha \in O_\Sigma$ **do**
6:         **if** $4 \leq \text{length}(\alpha) \leq 10$ **then**
7:             $\mathcal{M} \leftarrow \mathcal{M} \cup \{\alpha\}$ ▷ Add to the set of selected axioms
8:         **end if**
9:     **end for**
10: **end while**
11: $\mathcal{M} \leftarrow$ Randomly select 1000 axioms from $\mathcal{M}$.
12: Remove non-$\mathcal{EL}^{++}$ axioms from $\mathcal{M}$.
13: **return** $\mathcal{M}$

---

In detail, for each ontology $O$, we create a signature $\Sigma$ consisting of 1,000 randomly selected atomic concepts. Then, we perform forgetting over $O$ and $\Sigma$, resulting in a forgotten ontology $O_\Sigma$. Finally, we select all axioms in $O_\Sigma$ with a length[5] between 4 and 10. We repeat the progress multiple times if the generated axioms are less than 1000. The exact progress has been shown in Algorithm 1.

## E Issues with Axioms $A \sqcap B \sqsubseteq B'$

*Issues on box-based methods.* We identified a design flaw in the evaluation of axioms of the form $A \sqcap B \sqsubseteq B'$ in the Box2EL implementation. Specifically, the evaluation fails when $Box(A) \cap Box(B)$ is empty. The intersection box $Box(A \sqcap B)$ is always computed as follows:

(1) $\max(Box(A \sqcap B)) = \min\{c(Box(A)) + o(Box(A)), c(Box(B)) + o(Box(B))\}$;
(2) $\min(Box(A \sqcap B)) = \max\{c(Box(A)) - o(Box(A)), c(Box(B)) - o(Box(B))\}$;
(3) $c(Box(A \sqcap B)) = \frac{\max(Box(A \sqcap B)) + \min(Box(A \sqcap B))}{2}$;
(4) $o(Box(A \sqcap B)) = \frac{|\max(Box(A \sqcap B)) - \min(Box(A \sqcap B))|}{2}$.

This approach ignores cases where $Box(A) \cap Box(B) = \emptyset$, instead computing $o(Box(A \sqcap B))$ using the absolute difference between $\max(Box(A \sqcap B))$ and $\min(Box(A \sqcap B))$. This is neither logical nor coherent.

To address this, we refined the evaluation of axioms $A \sqcap B \sqsubseteq B'$ by setting $score(A \sqcap B \sqsubseteq B') = 0$ for all $B'$ when $Box(A) \cap Box(B) = \emptyset$. This is reasonable since $\bot \sqsubseteq C$ is always true for any $\mathcal{EL}$ concept $C$.

*Issure on ball-based methods.* Additionally, as the intersection of balls is generally not a ball, we cannot evaluate the performance of ELEM and ELEM++ on axioms like $A \sqcap B \sqsubseteq B'$. In [17], this issue was addressed by approximating the intersection of balls as boxes, using the same center, and defining the offset with all coordinates having the same value of the radii. However, we do not adopt this method, as it lacks a solid logical foundation.

## F Scoring function

Following [17], we define the scoring function for general concept inclusion axioms $C \sqsubseteq D$ using the distance between the centers of their Boxes. Formally,

$$s(C \sqsubseteq D) = -||\mathbf{c}(C) - \mathbf{c}(D)||.$$

Higher scores indicate a greater likelihood that $C \sqsubseteq D$ is a true axiom.

---

[5] We define the length of an axiom as the number of atomic concepts and roles it contains

## Table 11: Comparasion over axioms: $A \sqsubseteq ?B$

|  | Model | H@1 | H@10 | H@100 | Med | MRR | MR | AUC |
|---|---|---|---|---|---|---|---|---|
| GALEN | ELEm | 0.01 | 0.16 | 0.40 | 430 | 0.06 | 3568 | 0.85 |
| | EmEL++ | 0.02 | 0.16 | 0.37 | 632 | 0.06 | 3765 | 0.84 |
| | Box²EL | **0.04** | **0.30** | **0.51** | **89** | **0.12** | **2648** | **0.89** |
| | BoxEL | 0.00 | 0.00 | 0.05 | 3427 | 0.00 | 5656 | 0.76 |
| | ELBE | 0.02 | 0.12 | 0.28 | 751 | 0.06 | 3711 | 0.84 |
| | TransBox | 0.00 | 0.03 | 0.11 | 6518 | 0.01 | 8120 | 0.65 |
| GO | ELEm | 0.01 | 0.13 | 0.35 | 590 | 0.05 | 6433 | 0.86 |
| | EmEL++ | 0.01 | 0.12 | 0.30 | 1023 | 0.05 | 6709 | 0.85 |
| | Box²EL | **0.03** | **0.17** | **0.57** | **58** | **0.08** | **2705** | **0.94** |
| | BoxEL | 0.00 | 0.01 | 0.04 | 5594 | 0.00 | 13734 | 0.70 |
| | ELBE | 0.04 | 0.17 | 0.22 | 5098 | 0.08 | 9179 | 0.80 |
| | TransBox | 0.01 | 0.09 | 0.24 | 3076 | 0.04 | 9137 | 0.80 |
| Anatomy | ELEm | 0.07 | 0.30 | 0.57 | 43 | 0.14 | 9059 | 0.91 |
| | EmEL++ | 0.08 | 0.29 | 0.53 | 60 | 0.14 | 10414 | 0.90 |
| | Box²EL | **0.07** | **0.34** | **0.65** | **27** | **0.15** | **2918** | **0.97** |
| | BoxEL | 0.01 | 0.04 | 0.13 | 2109 | 0.02 | 10036 | 0.91 |
| | ELBE | 0.03 | 0.13 | 0.30 | 1353 | 0.06 | 11724 | 0.89 |
| | TransBox | 0.06 | 0.25 | 0.50 | 100 | 0.12 | 11575 | 0.89 |

## Table 12: Comparasion over axioms: $?A \sqsubseteq \exists r.B$

|  | Model | H@1 | H@10 | H@100 | Med | MRR | MR | AUC |
|---|---|---|---|---|---|---|---|---|
| GALEN | ELEm | 0.02 | 0.14 | 0.28 | 1479 | 0.05 | 4831 | 0.79 |
| | EmEL++ | 0.02 | 0.11 | 0.22 | 2240 | 0.05 | 5348 | 0.77 |
| | Box²EL | **0.08** | **0.18** | **0.32** | **662** | **0.12** | **3832** | **0.83** |
| | BoxEL | 0.00 | 0.02 | 0.07 | 7638 | 0.01 | 8792 | 0.62 |
| | ELBE | 0.00 | 0.07 | 0.13 | 5835 | 0.03 | 7623 | 0.67 |
| | TransBox | 0.00 | 0.08 | 0.19 | 4115 | 0.03 | 6597 | 0.72 |
| GO | ELEm | 0.06 | **0.40** | **0.52** | 54 | 0.15 | 6292 | 0.86 |
| | EmEL++ | 0.05 | 0.39 | 0.48 | 210 | 0.15 | 7788 | 0.83 |
| | Box²EL | 0.00 | 0.18 | 0.52 | 82 | 0.05 | 5085 | 0.89 |
| | BoxEL | 0.00 | 0.00 | 0.00 | 16735 | 0.00 | 18848 | 0.59 |
| | ELBE | 0.00 | 0.13 | 0.24 | 5092 | 0.03 | 11161 | 0.76 |
| | TransBox | 0.00 | 0.24 | 0.53 | 63 | 0.06 | 6808 | 0.85 |
| Anatomy | ELEm | 0.12 | 0.47 | 0.69 | 13 | 0.23 | 4686 | 0.96 |
| | EmEL++ | 0.13 | 0.42 | 0.60 | 23 | 0.23 | 7097 | 0.93 |
| | Box²EL | **0.21** | **0.56** | **0.75** | **7** | **0.33** | **2457** | **0.98** |
| | BoxEL | 0.04 | 0.17 | 0.33 | 885 | 0.08 | 12686 | 0.88 |
| | ELBE | 0.01 | 0.36 | 0.59 | 33 | 0.13 | 7667 | 0.93 |
| | TransBox | 0.02 | 0.42 | 0.70 | 17 | 0.14 | 6050 | 0.94 |

When extending this to boxes over $(\mathbb{R} \cup \{\emptyset\})^n$, the scoring function becomes:

$$s(C \sqsubseteq D) = -||(\mathbf{c}(C) - \mathbf{c}(D)) \cdot \mathbf{m}_C \cdot \mathbf{m}_D|| - M||\mathbf{m}_C \cdot (1 - \mathbf{m}_D)||,$$

where $M$ is a large constant. Recall that $\mathbf{m}_C$ and $\mathbf{m}_D$ are the masks of $Box(C)$ and $Box(D)$, indicating empty components along each coordinate. The second term ensures that $s(C \sqsubseteq D)$ becomes very small if, for any $i$-th coordinate, $Box(C)$ is non-empty (i.e., $\mathbf{m}_{C,i} = 1$) while $Box(D)$ is empty (i.e., $\mathbf{m}_{D,i} = 0$).

## Table 13: Comparasion over axioms: $\exists r.B \sqsubseteq ?A$

|  | Model | H@1 | H@10 | H@100 | Med | MRR | MR | AUC |
|---|---|---|---|---|---|---|---|---|
| GALEN | ELEm | 0.00 | 0.05 | 0.18 | 3855 | 0.02 | 6793 | 0.71 |
| | EmEL++ | 0.00 | 0.04 | 0.12 | 4458 | 0.01 | 7020 | 0.70 |
| | Box²EL | 0.00 | 0.07 | 0.16 | 4514 | 0.02 | 7317 | 0.68 |
| | BoxEL | 0.00 | **0.14** | **0.69** | **49** | **0.04** | **2869** | **0.88** |
| | ELBE | 0.00 | 0.00 | 0.01 | 11030 | 0.00 | 11139 | 0.52 |
| | TransBox | 0.00 | 0.01 | 0.07 | 7665 | 0.00 | 8835 | 0.62 |
| GO | ELEm | 0.01 | 0.49 | 0.60 | **12** | 0.12 | 6272 | 0.86 |
| | EmEL++ | 0.01 | 0.49 | 0.58 | **12** | 0.13 | 6442 | 0.86 |
| | Box²EL | 0.00 | 0.37 | **0.64** | 20 | 0.09 | **4971** | **0.89** |
| | BoxEL | 0.08 | 0.55 | 0.55 | 3492 | **0.27** | 10293 | 0.78 |
| | ELBE | 0.00 | 0.03 | 0.06 | 13218 | 0.01 | 15417 | 0.66 |
| | TransBox | 0.00 | 0.07 | 0.34 | 2096 | 0.02 | 9905 | 0.78 |
| Anatomy | ELEm | 0.00 | 0.03 | **0.23** | **813** | 0.01 | 10230 | 0.91 |
| | EmEL++ | 0.00 | 0.02 | 0.17 | 1470 | 0.01 | 10951 | 0.90 |
| | Box²EL | 0.00 | **0.05** | 0.15 | 2891 | **0.02** | **8284** | **0.92** |
| | BoxEL | 0.00 | 0.00 | 0.00 | 25795 | 0.00 | 30281 | 0.72 |
| | ELBE | 0.00 | 0.00 | 0.00 | 16239 | 0.00 | 26970 | 0.75 |
| | TransBox | 0.00 | 0.01 | 0.12 | 3562 | 0.01 | 12631 | 0.88 |

## Table 14: Ablation Study on GO.

| Task | SemEn | IntEn | H@1 | H@10 | H@100 | Med | MRR | MR | AUC |
|---|---|---|---|---|---|---|---|---|---|
| Complex | ✓ | ✓ | **0.16** | **0.41** | **0.66** | **30** | **0.25** | **730** | **0.95** |
| | | ✓ | 0.08 | 0.19 | 0.32 | 596 | 0.12 | 3143 | 0.77 |
| | ✓ | | 0.09 | 0.18 | 0.21 | 1052 | 0.12 | 4920 | 0.64 |
| | | | 0.02 | 0.04 | 0.07 | 1068 | 0.03 | 5184 | 0.62 |
| Normalized | ✓ | ✓ | 0.01 | **0.16** | **0.35** | **1390** | **0.06** | **8186** | **0.82** |
| | | ✓ | 0.00 | 0.06 | 0.16 | 5464 | 0.02 | 11733 | 0.74 |
| | ✓ | | 0.01 | **0.16** | 0.30 | 1498 | **0.06** | 9369 | 0.80 |
| | | | 0.01 | 0.07 | 0.15 | 4342 | 0.03 | 10925 | 0.76 |

## Table 15: Ablation Study on ANATOMY.

| Task | SemEn | IntEn | H@1 | H@10 | H@100 | Med | MRR | MR | AUC |
|---|---|---|---|---|---|---|---|---|---|
| Complex | ✓ | ✓ | **0.26** | **0.55** | **0.68** | **7** | **0.35** | **668** | **0.99** |
| | | ✓ | 0.01 | 0.02 | 0.05 | 989 | 0.01 | 29806 | 0.40 |
| | ✓ | | 0.17 | 0.22 | 0.24 | 1001 | 0.19 | 17421 | 0.65 |
| | | | 0.00 | 0.01 | 0.04 | 996 | 0.01 | 16560 | 0.66 |
| Normalized | ✓ | ✓ | **0.03** | **0.35** | **0.63** | 27 | **0.13** | **8238** | **0.92** |
| | | ✓ | 0.02 | 0.04 | 0.06 | 50045 | 0.02 | 49579 | 0.53 |
| | ✓ | | **0.03** | **0.35** | 0.61 | 30 | **0.13** | 8309 | **0.92** |
| | | | 0.01 | 0.04 | 0.06 | 49695 | 0.02 | 49280 | 0.54 |

Other methods, such as BoxEL [31], define a volume-based scoring function as:

$$s(C \sqsubseteq D) = -\frac{\mathrm{Vol}(C \sqcap D)}{\mathrm{Vol}(C)}.$$

## G  Other comparison results

The comparison results for other normalized axioms of the form $A \sqsubseteq B$, $A \sqsubseteq \exists r.B$, and $\exists r.B \sqsubseteq A$ are shown in Tables 11, 12, and 13. We use the ELEM and ELEM++ results from [17], as the error discussed in the previous section affects only the test set, not the training or evaluation sets. These results show that non-$\mathcal{EL}^{++}$-closed methods outperform in predicting normalized axioms. This is expected, as they are not constrained by the need to embed complex concepts, allowing better performance in capturing the semantics of normalized axioms.

The results of the ablation study for GO and ANATOMY are presented in Tables 14 and 15. It is evident that employing both Semantic Enhancement and Intersection Enhancement yields the best performance in both ontologies.

Furthermore, the impact of Semantic Enhancement and Intersection Enhancement varies between predicting Complex Axioms and Normalized Axioms. For instance, when predicting Normalized Axioms in GO, the AUC ranges from 74 to 82, whereas for Complex Axioms, it ranges from 62 to 95. This suggests a more pronounced effect on the latter task.

Overall, Semantic Enhancement generally improves model performance. However, the influence of Intersection Enhancement is relatively minor. In some cases, such as in the ANATOMY Ontology without Semantic Enhancement, it may slightly decrease performance.

