# OpenReview forum: "TransBox:  $\mathcal{EL}^{++}$-closed Ontology Embedding"
_ACM.org/TheWebConf/2025/Conference — WWW 2025 Oral_

### Official Review · Reviewer_6tuH · 2024-11-29

**Novelty:** 7
**Technical Quality:** 7

**Review:**

First, thank you for giving me an opportunity to review this paper. The authors propose the TransBox model to overcome the limitations of existing geometric-based models, such as their inability to satisfy $\mathcal{EL}$++ closure and represent many-to-many relationships. TransBox satisfies $\mathcal{EL}$++ closure and represents role embeddings as boxes, effectively modeling role inclusion and role composition. Furthermore, it flexibly handles one-to-many, many-to-one, and many-to-many roles, which TransE-based models fail to address.

### **Strengths**:

- The proposed model effectively addresses the limitations of existing models, with clear and sufficient theoretical proofs supporting its approach.
- Experimental results show that TransBox achieves significant performance improvements over state-of-the-art models, demonstrating high performance even on complex axioms compared to other models.

### **Weaknesses**:

- Section 6 appears to provide insufficient explanations for the experimental results.

**Questions:**

1. I could not find an explanation for the experimental results in Table 8. Could you confirm if the reference to 'Table 7' on line 864 is a typo and should instead refer to 'Table 8'?
2. Is the reason for not comparing Box2EL in Tables 3–6 due to Box2EL's inability to define conjunctions?
3. Is there a specific reason for excluding comparisons with the $\mathcal{ALC}$-ontology model in the experiments?

**Reviewer Confidence:**

2: The reviewer is willing to defend the evaluation, but it is likely that the reviewer did not understand parts of the paper

**Scope:**

4: The work is relevant to the Web and to the track, and is of broad interest to the community

---

### Official Review · Reviewer_Rdtx · 2024-12-01

**Novelty:** 6
**Technical Quality:** 6

**Review:**

The paper describes TransBox, an approach to generating ontology embeddings.  It is differentiated from existing approaches by its attention to the question of how geometric models of an $\mathcal{EL}^{++}$ ontology can be constructed to support the generation of embeddings for complex concepts through novel approaches to role composition and box intersections, doing so in a manner that is $\mathcal{EL}^{++}$-closed, which is desirable as it helps preserve logical constraints implicit in a given ontology. The paper clearly describes the motivation for these techniques, explaining them first using a small toy ontology, and then provides empirical results showing the efficacy of the approach relative to previous published approaches. This work is significant in that it suggests a way to improve the utility of ontology embeddings for the prediction of axioms and question answering given an ontology.

**Questions:**

The Introduction briefly addresses LM-based approaches that rely on textual information, and describe them as lacking with respect to adherence to logical constraints and transparency in the reasoning process. It would be useful to understand the author(s)' perspective on the relative merits of geometrical models vs LMs from the perspective of recent work on reasoning and logical consistency in frontier large language models. For example, are these techniques potentially complimentary? Are there aspects of the ontology learning, maintenance and question answering tasks that are better addressed using an LM-based or hybrid approach, or are geometric models indubitably superior across these use cases?

**Reviewer Confidence:**

2: The reviewer is willing to defend the evaluation, but it is likely that the reviewer did not understand parts of the paper

**Scope:**

3: The work is somewhat relevant to the Web and to the track, and is of narrow interest to a sub-community

---

### Official Review · Reviewer_GAG8 · 2024-12-03

**Novelty:** 3
**Technical Quality:** 6

**Review:**

The paper proposes TransBox, a novel EL++-closed ontology embedding method capable of handling complex Description Logic (DL) expressions. This approach enhances traditional ontology embeddings by representing both atomic and composite concepts through box-based embeddings, effectively addressing limitations in many-to-many and hierarchical relationships.

Strengths

1. The proposed TransBox model introduces advanced mechanisms for embedding EL++ ontologies, ensuring EL++-closure and role composition capabilities. These innovations contribute to robust reasoning tasks, including ontology learning and query answering.
2. Extensive experiments on real-world biomedical datasets (e.g., GALEN, GO, Anatomy) showcase state-of-the-art performance, especially for predicting complex axioms.
3. The soundness of the model is well-articulated, demonstrating that TransBox preserves logical consistency for ontology axioms.
4. The method is broadly applicable in domains requiring complex reasoning, such as healthcare and bioinformatics, and is supported by publicly available code and datasets.

Weaknesses

1. While comparisons with existing box-based models (e.g., BoxEL, ELBE) are thorough, the discussion of limitations relative to ball-based methods (e.g., ELEM, ELEM++) is relegated to the appendix. A direct comparison in the main text would strengthen the analysis.
2. While the method demonstrates robustness on benchmark datasets, the scalability to larger or more dynamic ontologies (e.g., real-time medical knowledge updates) is not extensively discussed.

**Questions:**

1. Is there a specific rationale behind selecting box-based embeddings over alternative representations, such as spheres or other geometric primitives, beyond EL++-closure?
2. Can the semantic enhancement method introduced in Section 4.3 be generalized to other types of description logics outside EL++?
3. Why is Box2EL, a relevant baseline mentioned in the paper, absent from the comparisons in Tables 2–6? Was this due to limitations in its applicability, initial performance issues, or other experimental considerations?

**Reviewer Confidence:**

3: The reviewer is confident but not certain that the evaluation is correct

**Scope:**

4: The work is relevant to the Web and to the track, and is of broad interest to the community